# Immune Signature of COVID-19: In-Depth Reasons and Consequences of the Cytokine Storm

**DOI:** 10.3390/ijms23094545

**Published:** 2022-04-20

**Authors:** Paulina Niedźwiedzka-Rystwej, Adam Majchrzak, Sara Kurkowska, Paulina Małkowska, Olga Sierawska, Rafał Hrynkiewicz, Miłosz Parczewski

**Affiliations:** 1Institute of Biology, University of Szczecin, 71-412 Szczecin, Poland; paulina.malkowska@phd.usz.edu.pl (P.M.); olga.sierawska@phd.usz.edu.pl (O.S.); rafal.hrynkiewicz@usz.edu.pl (R.H.); 2Department of Infectious, Tropical Diseases and Immune Deficiency, Pomeranian Medical University in Szczecin, 71-455 Szczecin, Poland; adammajchrzak@protonmail.com (A.M.); mparczewski@yahoo.co.uk (M.P.); 3Department of Nuclear Medicine, Pomeranian Medical University, 71-252 Szczecin, Poland; sarakurkowska95@gmail.com; 4Doctoral School, University of Szczecin, 71-412 Szczecin, Poland

**Keywords:** COVID-19, SARS-CoV-2, viral diseases, cytokines, cytokine storm, immune response, viral response, infection

## Abstract

In the beginning of the third year of the fight against COVID-19, the virus remains at least still one step ahead in the pandemic “war”. The key reasons are evolving lineages and mutations, resulting in an increase of transmissibility and ability to evade immune system. However, from the immunologic point of view, the cytokine storm (CS) remains a poorly understood and difficult to combat culprit of the extended number of in-hospital admissions and deaths. It is not fully clear whether the cytokine release is a harmful result of suppression of the immune system or a positive reaction necessary to clear the virus. To develop methods of appropriate treatment and therefore decrease the mortality of the so-called COVID-19-CS, we need to look deeply inside its pathogenesis, which is the purpose of this review.

## 1. Introduction

In January 2020, the World Health Organization (WHO) declared a pandemic caused by COVID-19 (coronavirus disease 2019), with waves continuing to spread across the globe. COVID-19 is a systemic disease with a wide spectrum of clinical manifestations induced by SARS-CoV-2 (severe acute respiratory syndrome coronavirus 2) [1,2]. The first pneumonia cases were identified in Wuhan in early December 2019 [3]. Globally, as of 9 February 2022, there have been 399,600,607 confirmed cases of COVID-19, including 5,757,562 deaths, reported to WHO [4]. Most of these cases are mild, but in some patients inflammation results in a hyperinflammatory syndrome associated with acute respiratory distress syndrome and end-organ damage. Much of the mortality has been associated with a cytokine storm (CS) syndrome in patients admitted to hospital with COVID-19 pneumonia [5]. Previous epidemics caused by SARS-CoV-1 and Middle Eastern respiratory syndrome-CoV were also characterized by a cytokine storm [6]. There are two forms of CS that exist in autoimmune diseases, malignancies, and infections. The first is hemophagocytic lymphohistiocytosis (HLH) and the second one is macrophage activation syndrome (MAS). Both rely on well-established criteria [5,7,8]. COVID-19-associated CS is a unique form of a hyperinflammatory response which has only recently been characterized in association with the SARS CoV-2 infection [9]. Cytokine storm, caused by the excessive secretion of cytokines, leads to cytokine release syndrome (CRS), a severe systemic inflammatory response [10]. By definition, the inflammatory response is designed to protect the host from damaging stimuli and is a mechanism necessary for recovery. However, an overactive inflammatory response, as in a cytokine storm syndrome, can cause widespread tissue damage and even death [11].

There are two positions on CS associated with COVID-19. One holds that the CS is characterized by an excessive and detrimental immune response which led to the idea of a maladaptive immune activity driving the morbidity and mortality associated with COVID-19 [12]. However, there are also critics of this idea who argue that suppression of the immune response to a microbial pathogen is contrary to decades of medical teaching and hypercytokinemia may be necessary to clear the virus [13]. The critical element in the fight against SARS-CoV-2 is effective treatment, however the efficacy of antivirals in the CS is limited. Understanding the pathogenesis and mechanisms occurring during infection are key to developing and refining the possible future treatment methods [2]. To this end, the current review emphasizes the duality of the cytokine storm phenomenon in COVID-19, which can be exploited in further studies.

Three states of disease severity are distinguished during SARS-CoV-2 infection. The first is mild COVID-19, which is characterized by fever, dry cough, fatigue, and abnormal chest CT findings, but with a good prognosis [14]. The clinical symptoms are mild and no pneumonia manifestation can be found in imaging [15]. The second group, moderate COVID-19, shows evidence of lower respiratory disease during clinical assessment or imaging with an oxygen saturation (SpO2) ≥94% in room air at sea level [16]. The last group, severe COVID-19, is characterized by increased cytokine production, lymphopenia, pneumonia, severe symptoms of acute respiratory distress syndrome (ARDS), multiple organ failure [17], tachypnea and dyspnea, higher neutrophil counts, and increased WBC [15], with a bad prognosis, and meeting any of the following: respiratory distress, respiratory rate ≥ 30 breaths/min; SpO2 ≤ 93% at rest; and PaO2/FIO2 ≤ 300. Patients with greater than 50% lesion progression within 24 to 48 h in pulmonary imaging should be treated as severe cases [15].

The World Health Organization’s (WHO) definition of severe pneumonia is categorized as severe disease. As of 27 May 2020, the WHO’s most recent clinical guidelines define “severe disease” as adults with clinical signs of pneumonia (fever, dyspnea, cough, and fast breathing) accompanied by one of the following: respiratory rate > 30 breaths/min; severe respiratory distress; or oxygen saturation (SpO2) ≤ 90% in room air. The precise determinants of severe disease are not known, but it appears that primarily host factors rather than viral genetic mutations drive the pathogenesis [18].

## 2. Virus Molecular Variability

Evolution of the SARS-CoV-2 variants, especially within the spike region, is leading to increased transmissibility and more effective evasion of the immune system. Evolution of SARS-CoV-2 has been monitored and assessed by WHO and other worldwide institutions since January 2020. Due to the increasing threat to global public health, new variants have been classified into groups, among which the most important from the current global public health perspective are Variants of Concern (VOCs) [19].

Well-established VOCs include Alpha, Beta, Gamma and Delta, responsible for various pandemic waves worldwide, with key characteristics related to mortality and transmissibility as shown in Figure 1.

On 25 November 2021, a new variant, now named Omicron, was reported in Botswana and South Africa and designated as a VOC almost immediately. The cause is believed to be high number of mutations in the spike glycoprotein, which most probably results in improvement of transmissibility, viral binding affinity and antibody escape, which may limit the vaccine efficacy. Currently, the severity of COVID-19 caused by Omicron is still hard to assess, especially in relation to the large scale of the expected infections. Nevertheless, preliminary studies report the reduced number of hospitalizations as well as decreased severity and mortality [20]. It was also previously suggested that some mutations in the novel variants may arise as a result of the non-human evolution of the virus. Recently published data suggest that the ancestor of Omicron might have host-jumped from humans to mice, to rapidly accumulate mutations specific to that host, and then jumped back into humans. This theory, however, should be regarded with caution until confirmed [21].

## 3. COVID-19 Transmission

Although SARS-CoV-2 human to human transmissions are vital for the evolution of the pandemic, the virus can also infect and replicate in an array of animal hosts, including companion, captive, wild, and farm animals, which may play a role as potential reservoir hosts for SARS-CoV-2 [22].

The household is an important site of SARS-CoV-2 transmission. Studies published by Z. J. Madewell et al. in January of 2020 report that the probability that an infection occurs among susceptible people within a specific group (SAR-secondary attack rates) were higher for SARS-CoV-2 vs SARS-CoV or MERS-CoV, household contacts vs other close contacts, adult vs child contact, spouses vs other contacts and symptomatic vs asymptomatic index cases [23]. A notable difference was also observed for the infecting variants. Moreover, further studies (August 2021), reported that estimated household SAR increased from 13.4% in 2020 to 18.9% in 2021, what is probably associated with more transmissible variants. It should also be emphasized that in-hospital infections remain common, where the risk of transmission is much higher, especially during procedures such as nebulization, endoscopy, endotracheal intubation, or cardiopulmonary resuscitation [24].

## 4. COVID-19 Clinical Manifestations

The clinical manifestations range from asymptomatic to life-threatening infection. Oran and Topol [25], based on the evaluation of 61 studies or reports, concluded that at least one-third of infections are asymptomatic. This result is consistent with the systematic review and meta-analysis performed by Sah et al. [26]. In a large study performed in China, Guan et al. [1] extracted data regarding 1099 hospitalized patients with laboratory-confirmed COVID-19. They analyzed patient data on many aspects of the disease, including their clinical course, radiographic and laboratory findings, complications, treatment, and outcome. The most common reported symptoms in this group of hospitalized patients were: fever (88.7% of patients), cough (67.8%), fatigue (38.1%), sputum production (33.7%), shortness of breath (18.7%). However, most of the patients present with mild to moderate symptoms of the disease, and, therefore, are treated in an ambulatory setting. The course of the disease of this group is less documented. According to Kim et al. [27], among 172 patients with mild presentation, the most common symptoms were cough (40.1%), hyposmia (39.5%), and sputum production (39.5%). Fever was only observed in 11.6% of cases.

Moreover, the clinical presentation among children differs from adults. De Souza et al. analyzed 38 studies, with a total of 1124 pediatric cases, and the most common symptoms were fever (47.5%), cough (41.5%), nasal symptoms (11.2%), diarrhea (8.1%), and nausea/vomiting (7.1%) [28]. The presence of comorbidities is also very important for the course of the disease. Barek et al. [29] investigated that relationship and they found out that the presence of the following comorbidities significantly increases the severity of symptoms: hypertension, diabetes, cerebrovascular diseases, cardiovascular diseases, respiratory diseases, malignancy, chronic kidney disease, and chronic liver diseases.

### 4.1. Respiratory Manifestations

The SARS-CoV-2 virus enters lung cells via the angiotensin-converting enzyme 2 (ACE2) receptor [30]. This mechanism is well studied and more details can be found in [31]. Human alveolar type II cells are the major target for SARS-CoV-2 virus [32]. Therefore, following fever, respiratory manifestations are the most prominent in patients affected by the symptomatic disease. They include sputum production, cough, dyspnea, pneumonia with acute respiratory distress syndrome (ARDS), and acute hypoxic respiratory failure [33].

ARDS is a severe complication, and it can also occur in the course of infections with other pathogens (bacteria, virus, fungus) with tropism for the respiratory tract, or during nonpulmonary sepsis, trauma, or aspiration. Regarding ARDS caused by SARS-Cov-2, renin-angiotensin system imbalance plays a crucial role [34].

It is difficult to estimate how many patients will develop serious complications such as ARDS or respiratory failure since available data are inconsistent and should account for the variant variability. For example, one of the first studies on that matter, from Wuhan, reported that 48.6% of patients hospitalized because of COVID developed ARDS [35]. Moreover, the group of patients with ARDS included older individuals with a higher number of comorbidities. In another study, Tzotzos et al. [36] considered seventeen cohorts from 2486 hospitalized COVID-19 patients, with the incidence of ARDS among hospitalized patients being 33%.

ARDS is undoubtedly associated with high mortality. In 2016, Bellani et al. [37] conducted an international, multicenter, prospective cohort study, based on which they concluded that the mortality of ARDS depends on its severity, and it is as follows: 34.9% for mild infections, 40.3% for moderate, and 46.1% for severe. Some authors state that the outcome of ARDS in the course of COVID-19 is worse compared to other etiologies [38]. Other authors, on the contrary, state that, in fact, ARDS in the course of COVID and non-COVID does not differ so much. For example, Ferrando et al. [39] in a multicenter, prospective, observational study analyzed 742 patients with ARDS and COVID-19. The 28-day mortality was 24% for mild, 29% for moderate and 32% for severe ARDS. Cytokine storm syndrome is one of the factors related to COVID-19 ARDS [40,41,42,43,44,45].

### 4.2. Extra-Respiratory Manifestations

The presence of ACE2 receptors in extra-pulmonary tissues and a tropism of SARS-CoV-2 to these receptors may lead to direct tissue and endothelial damage and dysregulation of local immune responses [46], which commonly lead to extra-respiratory disease manifestations. Lai et al. made a summary of the main extra-respiratory manifestations of patients with COVID-19, including cardiac, gastrointestinal, hepatic, renal, neurological, olfactory, gustatory, ocular, cutaneous, and hematological symptoms [47]. Among the most common ones were anosmia (79.6%), lymphopenia (56.5%), heart involvement (23–52%), and hepatitis (16.1–53.1%) [48].

Another serious manifestation associated with COVID-19 are thromboembolic episodes. In Italy, Lodigiani et al. [49] studied 388 hospitalized patients with COVID-19 in terms of venous and arterial thromboembolic complications. All patients in the group received thromboprophylaxis. They found out that thromboembolic events occurred in 7.7% of patients. The most common presentation was the venous thrombosis, most commonly pulmonary embolism (PE). In another study, conducted in the Netherlands, the authors reported 31% incidence of thromboembolic events among 184 patients in the intensive care unit who received thromboprophylaxis [50]. Similar to the previous report, PE was the most frequent thrombotic complication.

In general, the pathomechanism of venous thrombosis is explained by Virchow’s triad, which is altered blood flow, vascular injury, and hypercoagulability. It was found that SARS-CoV-2 has an impact on all three components of the triad. ACE2 receptors are present in endothelial cells, which allows SARS-CoV-2 endothelial invasion [51]. During autopsy examinations, it was found that SARS-CoV-2 infection causes endothelial dysfunction [52] which leads to loss of fibrinolytic function [53]. Maier et al. studied the viscosity of 15 critically ill patients with COVID-19, using capillary viscometry [54]. They found out that all of them had plasma viscosity exceeding 95% of normal.

Thromboembolic disorders are serious complication in COVID-19, especially during rapid interactions between inflammation, immunity, and the coagulation system, leading to cytokine storm, resulting in alveolitis, endothelitis, complement activation, recruitment of immune cells, as well as immunothrombosis. Despite standard thromboprohylaxis, the rate of venous thromboembolism is unusually frequent. An extremely important complication is disseminated intravascular coagulation (DIC), observed in 4.3-6.2% of COVID-19 patients, characterized by 26.2-times higher incidence of death [55,56].

Finally, SARS-CoV-2 causes a hypercoagulable state by (i) inhibition of the plasminogen system [53,57], (ii) platelet dysfunction [58,59], (iii) complement activation [60], (iv) hyperimmune response and thrombosis [61,62], and production of (v) antiphospholipid antibodies [63].

It is believed that the primary step of the cascade occurs in the lungs, where the severe inflammatory response triggers a dysfunctional cascade of inflammatory responses [64]. Firstly, it causes thrombosis in the pulmonary vasculature, and, then, it expands to the generalized condition. Ackermann et al. analyzed histologic samples of lungs of patients who died from COVID–19-associated respiratory failure, observing widespread thrombosis with microangiopathy [52]. Endothelial damage and subsequent coagulopathy are causative factors of the progression to severe manifestation. They can lead to disseminated intravascular coagulation and multiple organ failure, which results in death.

## 5. Cytokine Profile during SARS-CoV-2

Infection with Sars-CoV-2 virus results in the secretion of large amounts of inflammatory cytokines and chemokines. High levels of IL-2 (interleukin), IL-7, IL-10, G-CSF (granulocyte colony-stimulating factor), TNF (tumor necrosis factor), CXCL10 (CXC-chemokine ligand 10), MCP1 (monocyte chemoattractant protein-1), and MIP1α (macrophage inflammatory protein 1 alpha) in serum were observed in patients with severe COVID-19 (Figure 2) [3]. The secretion of such high cytokine levels indicates hyperactivity of the host immune system [65]. Interstitial mononuclear lymphocyte-dominated inflammatory infiltrates in the lungs and severe lymphopenia with hyperactive T cells in the peripheral blood are found in the postmortem pathology of patients with COVID-19. This is due to CS that further attracts immune cells, especially monocytes and T cells, causing pneumonia [66]. However, there is a lack of research on cytokine profiles in patients across the spectrum of COVID-19 disease severity. Most studies consider only mild and severe COVID-19, so these severity states are the focus of this chapter.

### 5.1. Signalling Pathways for Cytokine Storm in COVID-19

#### 5.1.1. SARS-CoV-2 Viral Entry

The S protein of SARS-CoV-2 virus binds to angiotensin-converting enzyme 2 (ACE2), which is its major receptor for most host cells [67]. ACE2 is highly expressed on alveolar cells, small intestinal enterocytes or vascular endothelium [51]. Another determinant of virus entry into the cell is the host serine protease (TMPRSS2), which stimulates the S protein to interact with the receptor. Studies show that pharmacological inhibition of TMPRSS2 prevents viral infection in animal models [68]. The S protein bridges the gap between the viral and host membrane cells by connecting them, resulting in the release of viral genomic RNA directly into the cytoplasm [67]. A second alternative pathway for SARS-CoV-2 entry into some host cells is internalization into endosomes and cathepsin-mediated cleavage triggered by low pH. After cleavage, viral membranes fuse with the endosomal membrane, which facilitates nucleocapsid entry into the cytoplasm [69].

The next step is translation of the SARS-CoV-2 genomic RNA into two large polyproteins (pp1a and pp1ab). These are responsible for encoding 16 non-structural proteins (NSPs) that facilitate the formation of a viral replication–transcription complex that generates an antisense negative-stranded matrix from viral RNA [67]. Subgenomic RNA is translated into structural and accessory proteins (S, M, E, and N). The structural proteins (S, M, E) are inserted into the ER and Golgi membranes and transverse to the ER–Golgi intermediate compartment (ERGIC). This mechanism results in the formation of virion-containing vesicles that can fuse with the plasma membrane during exocytosis. The fully formed virions are released into the extracellular space [70].

#### 5.1.2. Molecular Signaling Pathways Activated during Cytokine Storm in COVID-19

COVID-19 inflammatory signaling cascades are activated by receptors such as Toll-like receptors (TLRs), RIG-I-like receptors (RLRs), and NOD-like receptor that detect SARS-CoV-2 upon entry into the host cell [71]. Inflammatory pathways activated during SARS-CoV-2 infection include interleukin-6/Janus kinase/STAT (IL-6/JAK/STAT), interferon (IFN), tumor necrosis factor-α-nuclear factor-kappa (TNFα-NF-κB), and many others [72,73,74,75,76,77].

IL-6 and JAK/STAT signaling pathways are strongly associated with the COVID-19 CS. Pro-inflammatory cytokines such as TNF-α, IL-6 and IL-12 are produced inter alia by an array of immune cells such as lymphocytes, T lymphocytes, macrophages, dendritic cells or monocytes [78] as a result of the increased expression and activation of TLR7 and TLR8 in lung tissue [79].

IL-6 is an activator of the JAK/STAT3 pathway during inflammation. Studies from 2020 showed that the IL-6-JAK-STAT3 pathway is strongly associated with the severity of COVID-19 symptoms [80,81]. Higher levels of phosphorylated STAT3 could be observed in different leukocyte subsets in SARS-CoV-2 infected patients [82]. There are two signaling pathways that utilize IL-6, *cis* (classical) and *trans* (causing further activation of JAK/STAT3 signaling) [83]. During *cis* signaling, IL-6 binds to the membrane-bound IL-6 receptor (mIL-6R) and forms an IL-6/IL-6R/gp130 complex to activate JAK/STAT3, Akt/mTOR, and MAPK downstream signaling. mIL-6Rs undergo limited expression on the surface of immune cells, so formation of IL-6/IL-6R/gp130 complexes affects them. This is characterized by the promoted differentiation of type 17 helper T cells (Th17), CD8 + T cells and B cells, increased neutrophil migration and decreased development of Tregs [84]. In *trans* signaling, IL-6 binds to the soluble form of IL-6R (sIL-6R) to form a complex. This complex binds to the gp130 dimer, which is localized on almost all cell types. This allows the activation of IL-6-IL-6R-JAK-STAT3 pathway to occur without the involvement of mIL-6R [71]. The JAK/STAT pathway is also involved in Ang II and AT1R signaling which may increase the production of proinflammatory cytokines, whereas AT1R mediates the activation and phosphorylation of JAK2. The SARS-CoV-2 infection triggers inflammation via the JAK/STAT pathway leading to recruitment of pneumocytes, endothelial cells, macrophages, monocytes, lymphocytes, natural killer cells and dendritic cells progressing towards cytokine storm. This produces various inflammatory markers in the host that determine the disease severity [85]. The JAK/STAT signaling also mediates immune responses via B cell and T cell differentiation. Based on these data, it is suggested that JAK inhibition may play a key role in reducing cytokine production during SARS-CoV-2 infection [86].

It has been shown that IFN-γ is elevated in patients with COVID-19. IFN-γ is mainly produced by macrophages, T lymphocytes, and NK cells. It is responsible for activation of macrophages, NK cells, and T cells through activation of the JAK1/JAK2 complex [71]. Whether it plays an important role in the cytokine storm in COVID-19 is unknown; however, there is evidence that IFN-γ may be involved in CS-related disorders [87]. It may play a pathological role in primary hemophagocytic lymphohistiocytosis (HLH), during which there is a failure to eliminate pathogens due to impaired NK cell activity. Despite excessive T cell activation, elevated levels of IFN-γ result in defective NK cell activity [88]. However, in other infections such as HSV-2, the lack of IFN-γ increases viral replication and reduces survival [89].

TNF-α not only plays an important role in killing cancer cells, but it also has pleiotropic functions that include inflammatory response and host immunity to pathogens. Studies have shown that TNF-α can inhibit the replication of certain viruses. For example, TNF-α inhibits replication of human immunodeficiency virus type 1 (HIV-1) in peripheral blood monocytes and alveolar macrophages. However, it can stimulate replication of the same virus in chronically infected T lymphocytes and promonocyte cell lines [90]. TNFα/NF-κB signaling may be involved in the hyperactivation of the immune system that occurs during a cytokine storm. Numerous studies have shown that the above expression of TNFα represented a poor prognosis for patients infected with SARS-CoV and with MERS [91,92,93]. In contrast, NF-κB inhibition contributes to the improvement of pulmonary symptoms in SARS-CoV [94]. However, studies on COVID-19 showed that there was no correlation between TNFα expression and T-cell count and that TNFα was within normal limits [95,96]. SARS-CoV-2 infection is associated with increased NF-κB signaling, leading to antioxidant deprivation and inflammation [97]. The Nrf2 pathway is reported as a promising therapeutic target against COVID-19, as it and its antioxidant enzyme heme oxygenase-1 (HO-1) inhibit inflammatory regulatory pathways such as NF-κB [98]. Although a report has been published suggesting that inhibition of the TNFα/NF-κB signaling pathway has a protective effect for SARS-CoV-2 infection [99], it should be considered that its effect has not been fully elucidated and this may have negative side effects.

The NLRP3 protein forms a complex with apoptosis-related speck-like protein to cleave inactive IL-1β, a precursor to the mature form of IL-1β [100]. Coronavirus studies have suggested that IL-1β may contribute to the cytokine storm. Multiple cytokines with elevated levels have been identified in COVID-19, including IL-1β [101,102,103]. In addition, it has been shown that NLRP3 can be activated by SARS-CoV viral proteins (ORF3a and ORF8b) [104], which are also present in the SARS-CoV-2 genome [105], suggesting their potential similar role. A potential activator of NLRP3 is reactive oxygen species (ROS), so it is thought that their excessive production resulting from inflammation in COVID-19 leads to NLRP3 activation and cleavage of the IL-1β precursor which potentiates inflammation [106].

IL-2, through the IL-2R-JAK-STAT5 signaling pathway, induces differentiation of CD4 + T, CD8 + T, NK and other cells [71]. Its deficiency contributes to autoimmune diseases. Elevated levels of IL-2 have been observed during infections with other types of coronavirus [107,108]. These observations were also confirmed in SARS-CoV-2 infection, where both IL-2 and IL-2R were increased, especially in severe COVID-19 [15]. However, the role of IL-2 has not been fully understood, as other studies have shown that its plasma levels are not increased, but rather significantly decreased. In addition, significantly lower levels of JAK1 and STAT5 were noted. It has been suggested that decreased levels of IL-2, IL-2R, JAK1, and STAT5 are the reason why lymphopenia occurs with severe COVID-19 [109].

IL-7/IL-7R signaling is crucial for T cell differentiation (including CD4+, CD8+, naive T cells and memory T cells) [71]. IL-7/IL-7R signaling prevents apoptosis of CD4+ memory T cells by increasing anti-apoptotic protein levels through the JAK/STAT pathway [110]. Increased levels of IL-7 have been observed in patients with severe COVID-19, the cause of which is unknown. It is suggested that this may be a response to lymphopenia in patients with severe COVID-19 [96]. A similar phenomenon is observed in chronically HIV-1 infected individuals. IL-7 levels are elevated, whereas IL-7Ra expression is lower because IL-7 downregulates its own receptor [111].

IL-10 causes anti-inflammatory effects by transducing signals in the IL-10R/JAK/STAT3 pathway [112]. Studies on lymphocytic choriomeningitis virus (LCMV) infections have shown that IL-10 acts as an immunoregulator, inhibiting proinflammatory responses induced by innate and acquired immunity and preventing tissue damage due to an enhanced adaptive immune response [113]. However, patients with severe COVID-19 had significantly higher levels of IL-10 than patients with mild COVID-19 and healthy patients [114]. These results suggest that IL-10 may be a potential indicator of disease severity.

IL-12 signaling plays a key role in the cytokine storm as it increases immune cell activation. IL-12 is mainly produced by dendritic cells, macrophages and B lymphocytes [115]. In addition, it can promote proliferation of Th1 and Th17 cells and induce IFN-y expression in Th1 cells [116]. However, increased levels of IL-12 in SRAS-CoV-2 infection have not been reported [3], although they were seen in patients with SARS-CoV [42].

Granulocyte-monocyte stimulating factor (GM-CSF) is produced by endothelial, epithelial, hematopoietic, and other cell types. It is involved in the differentiation of macrophages, including alveolar macrophages (AM). In animal models of airway infection, intranasal administration of GM-CSF increased AM proliferation and improved outcomes [117]. However, studies have shown that increased levels of GM-CSF were seen in both mild and severe COVID-19 [3]. This has the adverse effect that in hyperinflammatory states (such as cytokine storm), GM-CSF leads to sudden myelopoiesis and perpetuates the inflammatory response by recruiting marrow cells to inflammatory sites [71].

### 5.2. Possible Consequences of Cytokine Storm in COVID-19

At the cellular level, in patients with severe COVID-19 (Figure 2), excessive release of pro-inflammatory cytokines leads to lymphopenia, lymphocyte dysfunction and granulocyte and monocyte anomalies (Figure 3) [17,118,119].

Lymphopenia occurs in the significant majority of patients with COVID-19 and its level is associated with a more severe disease course [3,66,71,120,121,122]. The development of a severe form of lymphopenia, a progressive decrease in lymphocytes, has been correlated with poor patient prognosis [58,123]. Remarkably, increased cytokine levels and lymphopenia were observed in patients with critical acute respiratory distress syndrome due to SARS-CoV in 2003 [124] and induction of apoptosis of primary T cells has been reported in MERS-CoV [125]. Lymphopenia can be induced by direct infection of SARS-CoV-2 T cells via the ACE2 receptor expressed on their surface causing their lysis and/or by increased numbers of regulatory T cells (Treg) [71]. In COVID-19 patients, damage to lymphocytes, CD4+T cells and especially CD8+T cells has been observed, involving a reduction in the number of lymphocytes in the peripheral blood and subsequent apoptosis [126]. Lymphocyte apoptosis may be associated with hypercytokinemia that may cause disruption of lymphocyte-producing organs and with a depletion phenotype [127,128]. Differences in lymphocyte subsets are observed in COVID-19 non-severe and severe cases. In severe cases, T cells (Th cells, memory Th cells, CD28-positive cytotoxic suppressor T cells, regulatory T cells) and NK cells are particularly vulnerable to destruction [120]. Increased activity of CD4 and CD8 T cells has been observed, with increased cytotoxicity of CD8 T cells and with an increase in the pro-inflammatory CCR6+ Th17 in CD4 T cells [66]. The association of lymphopenia with endothelial dysfunction in patients with organ failure and community acquired pneumonia (CAP) is indicated [129].

Lymphopenia and increased neutrophil counts were correlated with an increased risk of developing ARDS in COVID-19 patients and an overall more severe disease course [130,131,132,133]. Differences in neutrophil–lymphocyte ratio (NLR) in severe and non-severe patients are significant, with increased NLR in patients with severe COVID-19 [133,134,135]. A steady increase in neutrophil count, like a steady decrease in lymphocyte count, correlated with a poor prognosis for a patient with COVID-19 [135]. Similarly, a high NLR was a marker for a poor prognosis for a patient [136]. Inhibition of neutrophil apoptosis in infections has been shown to cause a significant increase in neutrophil apoptosis, resulting in increased cytokine and chemokine secretion [135]. Neutrophil extracellular traps (NETs) have been observed in patients with severe COVID-19 [137]. We reported on the role of neutrophils and NETs in the course of COVID-19 in our previous work [2].

Cytokine storm is a characteristic of macrophage activation syndrome (MAS), but in SARS-CoV-2 infection, macrophage parameters differ between those found in classical MAS [2]. There is no consistent benchmark for determining overall monocyte count in COVID-19 patients [138,139]. However, in COVID-19 lacked populations found in healthy individuals, such as CD14CD16 inflammatory monocytes, which were able to secrete GM-CSF [138], and FSC-high inflammatory monocytes with higher levels of the macrophage markers CD80 and CD206 [139]. In COVID-19 patients, an increased activity of the monocyte activation markers sCD14 and sCD163 was also observed [140]. Additionally, in COVID-19 patients, monocytes expressed a strong expression of the ACE2 receptor [139]. Strong monocytes induced IL-6, thus contributing to an increase in the severity of the cytokine storm [141]. Cytokine storm in COVID-19 patient is also associated with massive mononuclear cell infiltration in organs, thrombosis [142,143], and tissue hypoxia [144].

Macrophages are the most abundant type of immune cells in the lung under homeostatic conditions. Based on their location, they can be divided into at least two different populations, interstitial macrophages (IM) and alveolar macrophages (AM). IM are located in the parenchyma between the microvascular endothelium and alveolar epithelium, whereas AM reside in the lumen of the airspace and are in close contact with alveolar epithelial cells [145]. Some viruses, such as influenza, chikungunya, human herpes, and Zika virus, use monocytes and macrophages to replicate and spread the virus in the body [146,147,148,149,150,151]. Studies have shown that SARS-CoV also involves macrophages. However, in this case it did not lead to amplification of the virus, but only to impaired secretion of type I IFN [91].

In early COVID-19 disease, interferon (IFN) levels are reduced [152,153]. With its low levels and the onset of cytokine storm, the disease worsens [152]. This mechanism of IFN inhibition is important for SARS-CoV-2, not only to exacerbate the disease, but also because antiviral properties of interferons have been reported. These abilities have been demonstrated previously in studies of SARS-CoV [154,155] and inhibition of viral replication was also present in SARS-CoV-2 treated with IFN-I [156,157].

Higher cytokine levels were found in multiple studies. Unfortunately, it is uncertain whether the elevated levels of pro-inflammatory cytokines are reflecting damage or are protective [11]. Studies suggest that increased concentrations of certain cytokines/chemokines are dependent on SARS severity and can predict that the patient will progress to critical state or death [158]. Levels of IL-6, the cytokine that is most frequently studied in patients with COVID-19, are much lower than in classic ARDS [159,160].

Patients with severe COVID-19 present a similar cytokine profile to those with secondary hemophagocytic lymphohistiocytosis (sHLH), a serious hyperinflammatory syndrome caused by viral infections and involving uncontrolled cytokine secretion and hypercytokinemia [44,119]. HLH is divided into two types: primary and secondary. Primary occurs in the setting of a genetic predisposition that disrupts cytotoxic T cell and inflammasome function. It occurs more frequently in children. Secondary occurs more frequently in adults, but in the context of an immune trigger in an infection or malignancy. Both forms are triggered by activation of cytotoxic T cells and NK cells in combination with activation of macrophages [161]. HLH has been shown to occur predominantly in adults around age 50, and about two-thirds of cases are in men. It remains unclear whether certain races and ethnic groups are predisposed to developing HLH [162]. A close association between HLH and viral infections such as EBV, HIV, and CMV has been observed [163]. It has also been reported that SARS-CoV-2 leads to secondary HLH. It has not been fully explained why very few patients develop HLH after SARS-CoV-2 infection. It is speculated that the abnormal virus–host interaction is due to elevated cytokine levels, including IL-6 levels. IL-6 is a major driver of B-cell hyperactivity, leading to polyclonal expansion, and its excessive release is associated with numerous autoimmune diseases [84].

The onset of a cytokine storm resembles Systemic Inflammatory Response Syndrome (SIRS) [164] with an imbalance between pro-inflammatory and anti-inflammatory responses [165]. COVID-CS is largely responsible for multiple organ dysfunction syndrome (MODS) of which ARDS and/or SIRS are major components [166]. MODS is led by SARS-CoV-2 infection of multiple organs due to the virus entering the blood and spreading throughout the body [167]. Between the diagnosis of MODS in patients with COVID-19, there are 5–7 days of time when the syndrome develops [168]. This is a potential time period when therapies can be introduced. IL-6 and IL-1 are also significant activators of CSS and the inflammatory cascade [169]. Cytokine storm can lead to alveolar structural damage and lung ventilation dysfunction by damaging the lung capillary mucosa and by promoting alveolar edema [170].

The consequences of cytokine storm vary in patients depending on individual factors. Increased risk of developing a severe condition occurs in male gender, increased patient age and having at least one kind of chronic disease [171,172,173,174]. In female COVID-19 patients, the expression of the inflammatory cytokines IL-17, IL-22, and IL-3 was lower than in male patients. This was a characteristic feature, and its uniqueness was confirmed by its unrelatedness to patient age. Overall, men have an increase in cytokine and chemokine expression in COVID-19, and this may be an important factor in the severity of the disease [171]. The reason for this difference among males and females is still unknown. The effect of IFN on cytokine regulation seems to be an interesting hypothesis. As we mentioned earlier, IFN inhibition early in the disease promotes the development of a cytokine storm. In women, there is increased activation of IFN pathways early in SARS-CoV-2 infection [175,176], which may influence their lower cytokine expression. When considering the importance of gender in SARS-CoV-2 infection, it is important to note that females in general are more resistant to viral infections and have a greater diversity of immune responses [175]. Furthermore, it is important to mention how hormones affect the regulation of cytokine production. An important factor regulating cytokine production is estrogen. Estrogen inhibits IL-2 cytokine production and increases IFN-γ production; its high concentration promotes IL-10 regulatory cytokine production, and inhibits pro-inflammatory cytokines (IL-12, IL-6, and IL-1β), whereas its low concentration enhances their production [175]. Testosterone decreases the levels of IL-1β, IL-6, and TNF-α. In a mouse model, testosterone decreased cytokines, including IFN-γ and TNF, but also increased regulatory cytokines such as IL-10. Additionally, in mice, decreased production of the proinflammatory cytokines IL-1β and TNF is associated with the presence of progesterone [175].

Vitamin D3 is increasingly being identified as one of the factors influencing the course of COVID-19, including the development of CS. Vitamin D3 regulates the cytokine response to pathogens through more than one pathway, thereby affecting macrophage defense against viral pathogens [177]. Additionally, it promotes the production of anti-inflammatory cytokines and inhibits the production of pro-inflammatory (including TNF-alpha and IL-6) cytokines in lymphocytes [177,178]. It has been shown that its regulatory effects are stronger in women than in men [177]. This may be related to its estrogen-dependent effects controlling T-lymphocyte differentiation [178]. Vitamin D3 deficiency is also associated with skin color and where the patient lives. In Irish people over the age of 60, vitamin D deficiency correlated with an increase in IL-6 levels [179]. People living in countries in the northern hemisphere with low ultraviolet B levels had higher COVID-19 mortality correlated with increasing latitude north of 28 degrees north [177]. In patients with dark skin color, vitamin D deficiency is 8 times more common than in patients with light skin color. Black people had a mortality rate 3.6 times higher than white people, and the under-65 group without comorbidities had a mortality rate 12.6 times higher [180]. Williamson et al. [181] also showed increased mortality among Black people and South Asians [181]. There are too little data to determine the primary reason for the increased COVID-19 mortality in Black people. In addition to factors shared with white people, differences in socio-demographic indicators are also highlighted in Black people [182].

In addition to gender and race, vitamin D deficiency may also be associated with obesity. Reduced levels of the vitamin D biomarker, 25-hydroxyvitamin D, have been observed in obese patients too [178], which may also affect the dysregulation of cytokine levels. Obesity is a constant state of inflammation in the patient. It is characterized by a constant release of cytokines (including TNF-α, IL-6, and IL-1) by the patient’s adipose tissue [183]. Adipokines, adipose tissue-specific cytokines such as leptin, adiponectin, resistin, and visfatin, are also secreted. These adipokines may regulate the secretion of inflammatory and anti-inflammatory cytokines [183], and activation by proinflammatory cytokines of two signaling pathways, NF-kB and JNK, is characteristic of obesity [184]. Furthermore, leptin in particular can deregulate pro-inflammatory cytokines level in obesity [183]. As a consequence of this uncontrolled cytokine regulation, obese patients have consistently elevated levels of proinflammatory cytokines, which may result in increased susceptibility in obese patients to develop CS during COVID-19 [185]. The presence of obesity in a patient increases the chances of developing CS, resulting in ARDS, MODS, and death [186]. Additionally, visceral obesity has been suggested as a risk factor for severe COVID-19 [184].

### 5.3. Cytokine Storm as a Potential Biomarker in COVID-19 Progression

Cytokine profile analysis has been suggested as a potential biomarker for the development of COVID-19 and its associated superinfections, lymphopenias, ARDS, SIRS, and MODS [187,188,189,190,191], and can also be used for risk and prognosis assessment and analysis of response to therapies [190]. To date, cytokine storm has been used as a biomarker for viral diseases such as influenza or MERS [168]. It was reported that a high concentration of IL-6, IL-8, and MCP-1 may be a marker for bacterial or fungal superinfection [158], which is a serious problem in patients infected with SARS-CoV-2 which results in therapeutic and diagnostic challenge [192].

Furthermore, levels of IFN, IL, and TNF have been suggested as possible biomarkers of COVID-19 severity status [187]. Geppert et al. [193] found that in patients with cardiogenic shock, IL-6 serum levels can be used as a sensitive marker for the occurrence of MODS, as they are related to its severity. Furthermore, elevated IL-6 levels are associated with poor COVID-19 patient prognosis [189,194]. Correlation of poor prognosis has also been reported with the pneumonia cytokines IFN-γ and IL-10 [195].

A meta-analysis by Melo et al. [194] including 40 studies and 9524 patients with COVID-19 showed that there are substances that can be biomarkers for CSS. Eljaaly et al. [196] in their work suggested creating a uniform definition of CS in patients with COVID-19 so that a range could be created for cytokines as biomarkers of disease progression.

## 6. Biologic Drugs Used in the Treatment of COVID-CS

For a long time, traditional anti-inflammatory drugs were used to treat cytokine storm. However, biologic drugs involving inhibition of specific cytokines, or signaling pathways, and recombinant cytokines to counter inflammation, have been in development for some time (Figure 4).

### 6.1. Recombinant Cytokines

Previously, IFN-I has been used as one of the treatments for SARS-CoV and MERS-CoV infection. However, its efficacy in humans has not been clearly confirmed [197]. A study on COVID-19 reported that administration of low concentrations of IFN-α and IFN-β reduces viral titers and inhibits viral replication in host cells. In addition, SARS-CoV-2 was found to be more sensitive to IFN-I than SARS-CoV [198]. Subsequent studies have confirmed that the use of IFN-α2b alone or in combination with arbidol accelerates viral clearance and enables the restoration of normal IL-6 and CRP levels in patients infected with SARS-CoV-2 [199]. The successful use of recombinant IFN-α in the form of nasal drops as a preventive measure was also an important finding. Individuals in epidemic areas receiving the drops for 30 days showed a zero incidence rate [200]. In addition, the antiviral effects of IFN-III (pegylated IFN-λ1a) were tentatively confirmed in human upper respiratory tract epithelial cells and in SARS-CoV-2 infected mouse models [201].

Another cytokine that showed beneficial effects when treating severe COVID-19 was IL-7. Administration of recombinant human IL-7 (rhIL-7) to a patient with severe COVID-19 resulted in improved lymphocyte function, increased lymphocyte counts, and higher mHLA-DR expression [202]. In another study, administration of CYT107 (a commercial rhIL-7 product) resulted in clinical improvement in patients with severe COVID-19. It is suggested that administration of IL-7 in combination with dexamethasone may be the optimal treatment for COVID-19 because dexamethasone increases IL-7 activity by increasing expression of its receptor (IL-7Rα) [203].

### 6.2. Inhibition of Selected Cytokines

The damaging effects of cytokine storms on the body were the reason to search for solutions to decrease cytokine levels. The outcome of this research was the revelation that cytokine-lowering methods have a positive effect on COVID-19 treatment. There are several methods that we can use to block evolution of cytokine storm in COVID-19. The first is associated with interleukin-6. Monoclonal antibodies, competitive to interleukin-6-receptors, bind to receptors and therefore prevent it from interacting with transmembrane glycoprotein 130 (gp130), blocking intracellular signal transduction [204]. The main representative of this group is the recombinant humanized antibody, named tocilizumab, that inhibits interleukin-6 signaling by targeting both soluble (sIL-6R) and membrane (mIL-6R) forms of the interleukin-6 receptor. Although it is mostly used in rheumatoid arthritis and CAR-T (chimeric antigen receptor T cell) therapy, several studies confirm its efficacy in the treatment of severe COVID-19, due to the ability to improve blood oxygenation, reduce oxygen dependence, and limit radiological lung changes [205,206,207,208,209,210,211]. Systemic reviews and meta-analyses have assessed the efficacy of tocilizumab for the treatment of severe COVID-19 and confirm lower mortality in groups treated with tocilizumab than not treated, on moderate and critically ill patients (52.9 vs. 76.7%, respectively) [212]. Another study of IL-6R inhibition with sarilumab showed a reduction in CRP levels and improvement in clinical symptoms in patients with COVID-19 [213], while patients with severe COVID-19 requiring ventilator support who received siltuximab (an IL-6 inhibitor) had a significantly reduced mortality rate compared to patients who did not receive this treatment [214]. It is suggested that other IL-6 inhibitors (clazakizumab, sirukumab, olokizumab) and IL-6R (levilimab) should also be investigated as they may also prove to be potentially good therapeutic agents during the cytokine storm in COVID-19 [71].

The next group are recombinants of natural IL-1 receptor antagonist (IL-1Ra), which blocks the biological activity of proinflammatory cytokines (IL-1α and IL-1β), by competitive inhibition of IL-1 binding to its receptor (IL-1-R1). Their best known representant, anakinra, is officially approved for treatment of rheumatoid arthritis and autoinflammatory disorders, but is also used in septic shock and macrophage activation syndrome. In COVID-19 it has been proved to significantly decrease overall mortality, as well as need for mechanical ventilation, with no difference in adverse events [215]. Another study on the IL-1β antagonist canakinumab showed that it rapidly reduces inflammation and improves oxygenation in patients [216].

Another monoclonal antibody approved for the treatment of primary HLH is emapalumab. It is an IFN-γ inhibitor and its combination with anakinra has been shown to alleviate hyperinflammation and improve respiratory conditions [71].

Ethanercept, which is a soluble TNFα receptor fusion protein, can also be used during SARS-CoV-2 infection. A case of SARS-CoV-2 infection after subcutaneous treatment with etanercept for spondyloarthropathy showed no signs of hyperinflammation or respiratory failure [217]. In addition, rapid recovery from COVID-19 was reported. In contrast, infliximab and adalimumab are TNFα inhibitors, so their potential therapeutic effects are also under investigation [71].

Another interesting way to restrain the cytokine storm might be found in the usage of melatonin, used widely in sleep disorders, but its positive effect has been proved also against acute respiratory stress caused by many infections. Its beneficial effects result from indirect antiviral actions due the impact on pro-inflammatory cytokines (IL-1-beta, IL-2, IL-6, IL-8, IL-10, IFN-y, TNF-a, VEGF), oxidation stress (ROS and oxidative enzymes), and immune response (neutrophils, lymphocytes, CD8+ cells) [218].

The advantage of using melatonin in COVID-19 has been evaluated in some randomized controlled trials, confirming that patients receiving it have developed thrombosis and sepsis significantly less frequently during the second week of infection and had lower mortality rate than patients in the control group [219,220].

Based on available research, a supplementary use of melatonin might lead to a significant reduction of TNF-α and IL-6 levels, and can be an advisable supplement to current treatment in cytokine storm caused by COVID-19.

Effects of IL-12/IL-23 and IL-17A inhibitors have been observed in patients with psoriasis and COVID-19. Although risankizumab, guselkumab, tildrakizumab (targeting IL-23p19), and ustekinumab (targeting IL-12/IL-23p40) are widely used in the clinical treatment of chronic infectious and autoimmune diseases [221,222], case reports of their use appear to be poorly described [223,224,225,226]. Further research in this area will be needed to begin their use in the treatment of COVID-19. In contrast, inhibition of IL-17A expression has greater potential for use. It has been shown that patients treated for psoriasis with secukinumab and ixekizumab showed mild symptoms of COVID-19 or were asymptomatic [227,228]. Additional studies should be conducted in this regard as well.

Studies in patients with severe COVID-19 have shown that mavrilimumab (a GM-CSF-Rα inhibitor) can be used to improve patients’ condition earlier and reduce the progression to mechanical ventilation [229]. Lenzilumab, a recombinant monoclonal antibody against human GM-CSF, also has the potential to be used to treat COVID-19. Patients who received intravenous lenzilumab treatment had reduced progression to ARDS and reduced inflammatory markers [230]. Furthermore, GM-CSF inhibitors such as gimsilumab, otilimab, and TJ003234 are also being studied.

### 6.3. Blocking Signaling Pathways

The second group is based on the blockage of Janus kinase signal transducer and activator of transcription (JAK-STAT) pathway, which plays a pivotal role in signaling of Ang II and proinflammatory cytokines (TNF-alpha, IL-1, IL-6. IFN-gamma) in peripheral tissues and immune cells. JAK inhibitors, currently used in many illnesses such as rheumatoid arthritis, myeloproliferative neoplasms or inflammatory bowel disease, have been proved to reduce clinical symptoms in lungs, kidneys, and heart (by blockage of AT-1R), as well as confine the release of cytokines in COVID-19-CS and ARDS [86,231].

Baricitinib is a selective Janus kinase inhibitor that has proved its efficacy in patients with autoimmune diseases. After being identified by an artificial intelligence platform as a potentially useful intervention for the treatment of COVID-19 in February 2020 [232], several further studies confirmed its biochemical inhibitory effects responsible for SARS-CoV-2 viral propagation [233]. Marconi V. et al. conducted extensive research, gathering 1518 patients from 101 centers across 12 countries, in which patients treated with addition of baricitinib to standard treatment were associated with reduced mortality to that of standard care, with similar safety profile [234]. Last but not least, Kalil et al. [235] reported that treatment with both baricitinib and remdesivir versus remdesivir alone in patients with COVID-19 reduced recovery time, accelerated clinical improvement, and caused fewer serious adverse events.

It is worthwhile to mention that both IL-6-R and JAK-STAT pathway inhibitors should always be given simultaneously with dexamethasone or another corticosteroid. Drugs of choice in these groups are tocilizumab and baracitinib, but if they are not available or not feasible to use, tofacitinib and sarilumab may be used instead [236].

In addition, studies have shown that the use of ruxolitinib in the treatment of COVID-19 contributes to the reduction of seven cytokines and regeneration of lymphocyte counts [237].

For the NF-κB signaling pathway, several well-known antiviral and anti-inflammatory drugs such as dexamethasone, and hydroxychloroquine are suggested. Their use suppresses the NF-κB signaling cascade which reduces the expression of IL-6, TNFα, and IL-1β, among others [71,238,239].

Inhibition of NLRP3 signaling may be another possible treatment for COVID-19. Selective inhibitors such as dapansutrile and ZYIL1 block NLRP3 activation [71,240]. In addition, it has been shown that colchicine can non-selectively inhibit NLRP3 inflammasome by inhibiting P2X7 receptor activation or the interaction between NLRP3 protein and ASC [241].

## 7. Conclusions

COVID-CS is a unique phenomenon that is unwanted in a patient infected with SARS-CoV-2, but, on the other hand, its presence can help both in early diagnosis and in adjusting appropriate therapy. As usual, the intricacies of this biological process are not one-dimensional. Nevertheless, the widened knowledge on the processes in COVID-19 may contribute to adjusting treatment perspective depending on the patient’s condition. In our understanding, a good idea for further development of an effective treatment might be standardizing the immunological profile of CS in patients with COVID-19, in order to get a better diagnostic idea of the issue.

## Figures and Tables

**Figure 1 ijms-23-04545-f001:**
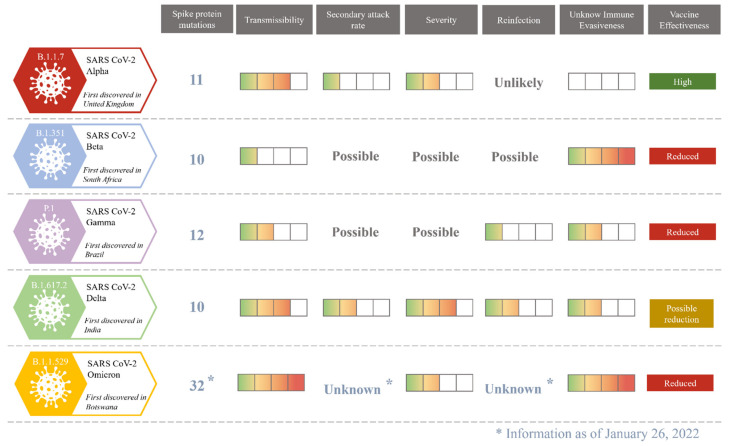
The comparison between currently known SARS-CoV-2 variants of concern. (Spike protein mutations—the number of mutations detected in protein S; Transmissibility—the severity of virus’s infectivity; Secondary attack rate—the probability of the infection among a specific group of people after contact with an infectious person in a household or other close-contact environment; Severity—the probability of severe course of COVID-19; Reinfection—the probability of a second or further development of COVID-19; Unknown Immune Evasiveness—the risk of the virus’s ability to escape neutralization; Vaccine effectiveness—the efficiency of currently investigated vaccines against SARS-CoV-2).

**Figure 2 ijms-23-04545-f002:**
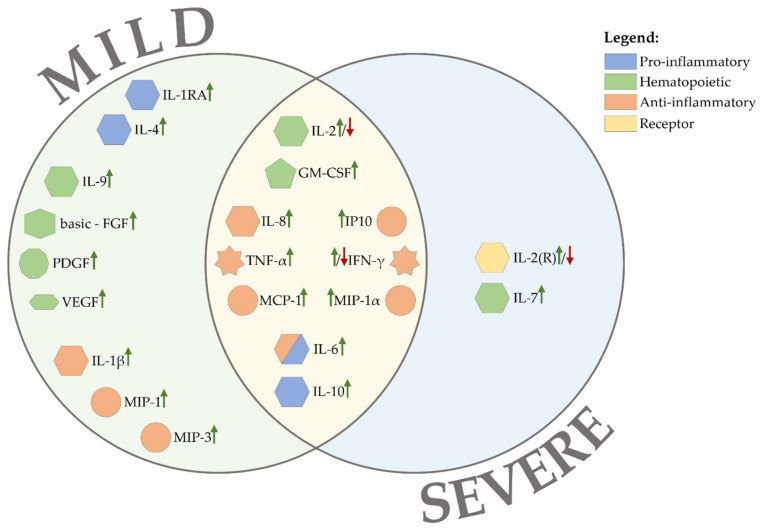
Cytokine levels change in COVID-19 patients depending on the patient’s condition.

**Figure 3 ijms-23-04545-f003:**
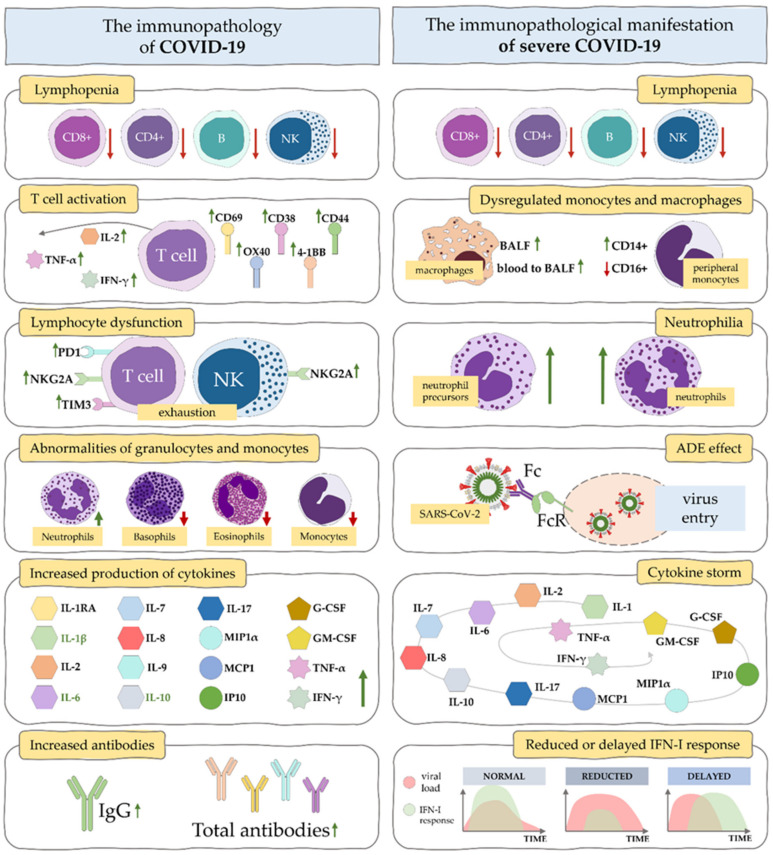
The immunopathology of mild and moderate COVID-19 and immunopathological manifestation of severe COVID-19.

**Figure 4 ijms-23-04545-f004:**
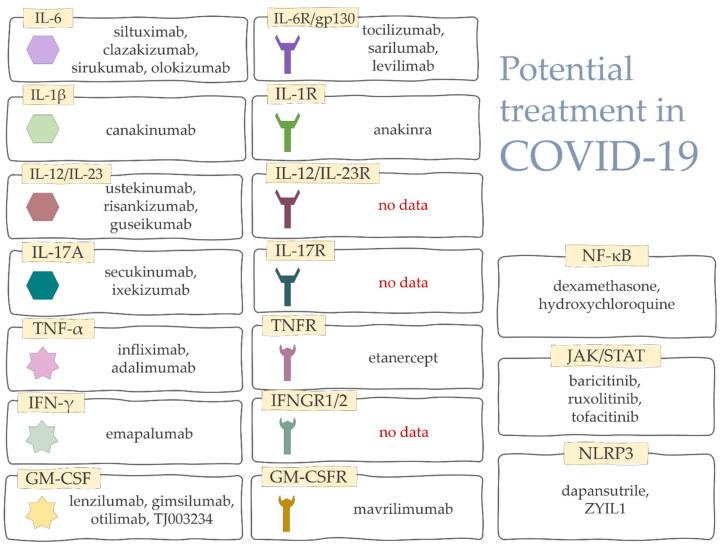
Main representatives of drugs used in COVID-19-CS.

## Data Availability

Not applicable.

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
