# Peer review of "Immune Signature of COVID-19: In-Depth Reasons and Consequences of the Cytokine Storm"

_ijms, 2022, doi:10.3390/ijms23094545_

Round 1

Reviewer 1 Report

Congratulations, your work is very interest and actual!

Author Response

Dear Reviewer,

We appreciate the time and effort that you have dedicated to providing your valuable feedback on my manuscript. Thank you very much for your positive review of our publication. We are glad that we met all your expectations.

Kind regards,

Paulina Niedźwiedzka-Rystwej

Reviewer 2 Report

The review is well set up and addresses a crucial problem about the complications of Covid-19 infection,  related to the presence of CS. The review is developed starting from basic science concepts, and with appropriate correlations with the clinic reports.

The discussion, actually, tackles and updates the problem of variants and their characterization.

The title of the review, however, emphasizes the pathological role of CS which, however, is re-expressed in a limited way, except in the introduction. I therefore think that an in-depth study is appropriate; at the same time in the characterization of HLH and MAS it would be advisable to provide more clarifications on these expansions of the CS in the different settings; with reference to the documented pictures of myocarditis, the detection of intramyocytic viruses is rare, while in the literature there is evidence of its presence within macrophages.

Furthermore, even considering the uncertainties concerning the topic in general, it would be appropriate to correlate the findings of clinical cases dominated by CS in the various and subsequent pandemic waves that followed from winter 2019-2020.

Author Response

Dear Reviewer,

We appreciate the time and effort that you have dedicated to providing your valuable feedback on our manuscript. We are grateful for your insightful comments on our paper. We have been able to incorporate changes to reflect most of the suggestions. Here is a point-by-point response to the reviewer's comments and concerns.

Review 2

The review is well set up and addresses a crucial problem about the complications of Covid-19 infection,  related to the presence of CS. The review is developed starting from basic science concepts, and with appropriate correlations with the clinic reports.

The discussion, actually, tackles and updates the problem of variants and their characterization.

  1. The title of the review, however, emphasizes the pathological role of CS which, however, is re-expressed in a limited way, except in the introduction. I therefore think that an in-depth study is appropriate; at the same time in the characterization of HLH and MAS it would be advisable to provide more clarifications on these expansions of the CS in the different settings; with reference to the documented pictures of myocarditis, the detection of intramyocytic viruses is rare, while in the literature there is evidence of its presence within macrophages.

We have completed the information on HLH and MAS. In addition, we added a paragraph in the "Possible consequences of cytokine storm in COVID-19" section regarding the possibility of detecting the presence of the virus in macrophages

  1. Furthermore, even considering the uncertainties concerning the topic in general, it would be appropriate to correlate the findings of clinical cases dominated by CS in the various and subsequent pandemic waves that followed from winter 2019-2020.

Unfortunately, looking for a correlation between epidemic waves and the occurrence of cytokine storm seems to be impossible. The authors make no reference in their paper to which epidemic wave the patients came from. Moreover, we believe that the reference to the wave is not that important and the viral variant seems to be more relevant. Regarding variants, we have referred to the "Virus molecular variability" section; however, please note that in the cytokine storm papers, the authors refer to SARS-CoV-2 in general without citing specific variants. However, your suggestion is very interesting and we will certainly include the virus variant in our future studies.

Again, we would like to thank you for your time, expertise, and effort in correcting our paper, which improved due to the changes you have proposed. We hope that now it fulfils the requirements to be published.

Kind regards,

Paulina Niedźwiedzka-Rystwej

Reviewer 3 Report

Review:

Immune signature of COVID-19 – in-depth reasons and consequences of the cytokine storm.

In this review the authors focus on understanding the COVID-19 cytokine storm pathogenesis in order to develop ways of appropriate treatment and consequently decrease this syndrome’s mortality.

Minor observations:

Abstract

Add a to the following paragraph:

From:  It is not fully elucidated, if the cytokine release is harmful result of suppression of immune system or positive reaction necessary to clear the virus.

To: It is not fully elucidated, if the cytokine release is a harmful result of suppression of immune system or positive reaction necessary to clear the virus.

A Slight discrepancy in figure 1: In the plot, “Unknown immune evasiveness” it is indicated, while in the legend “Unknown immune evasion” is written.

Major observations:

It is necessary to be more explicit about the Cytokine Storm in terms of the function type of cytokines and if they are up or downregulated under a particular circumstance. This will provide readers with a first glance of the processes that could be activated depending on the patient’s condition.

I recommend modifying figure 2 in terms of cytokines functions (proinflammatory, hematopoietic, etc.) to better understand the idea of mild and severe condition because this is a very fuzzy categorization.

There is an important clinical manifestation that is necessary to be included in the review, to provide a holistic view of the disease. It is necessary to focus part of the discussion on how population diversity impacts acute respiratory distress syndrome. Some questions that are important to address are:

Can these differences in mortality rate be attributed to differences in populations?

Is there any difference in race-ethnicity between studies?

What is the average age of the individuals from one study vs the other?

In other words, a more rigorous comparison among studies is necessary so the results can be compared.

Another important aspect that must be included in the review is the key role of sex and age on cytokine synthesis after COVID-19 infection (https://www.ncbi.nlm.nih.gov/pmc/articles/PMC7761414/).

There is a significant difference in the way males and females face COVID-19 infection and it has to do with the proportion of cytokines and chemokines. I strongly suggest reading about these differences because this can help you explain the differences in mortality rates reported in other articles.

In conclusion, the review is interesting, but it is missing of deeper discussion and conclusions from the authors. A review requires that the authors not only compile information but also, reach to interesting insights, formulate hypotheses, and propose interesting ideas for future directions.

Author Response

Dear Reviewer,

We appreciate the time and effort that you have dedicated to providing your valuable feedback on our manuscript. We are grateful for your insightful comments on our paper. We have been able to incorporate changes to reflect most of the suggestions provided. Here is a point-by-point response to the reviewer's comments and concerns.

Review 3

Immune signature of COVID-19 – in-depth reasons and consequences of the cytokine storm.

In this review the authors focus on understanding the COVID-19 cytokine storm pathogenesis in order to develop ways of appropriate treatment and consequently decrease this syndrome’s mortality.

Minor observations:

Abstract

Add a to the following paragraph:

From:  It is not fully elucidated, if the cytokine release is harmful result of suppression of immune system or positive reaction necessary to clear the virus.

To: It is not fully elucidated, if the cytokine release is a harmful result of suppression of immune system or positive reaction necessary to clear the virus.

A Slight discrepancy in figure 1: In the plot, “Unknown immune evasiveness” it is indicated, while in the legend “Unknown immune evasion” is written.

Thank you for your feedback, minor mistakes have been corrected

Major observations:

  1. It is necessary to be more explicit about the Cytokine Storm in terms of the function type of cytokines and if they are up or downregulated under a particular circumstance. This will provide readers with a first glance of the processes that could be activated depending on the patient’s condition.

In "Molecular signaling pathways activated during cytokine storm in COVID-19," we described the functions of individual cytokines and signaling pathways and determined whether they are up- or down-regulated.

  1. I recommend modifying figure 2 in terms of cytokines functions (proinflammatory, hematopoietic, etc.) to better understand the idea of mild and severe condition because this is a very fuzzy categorization.

 As requested, we divided the listed cytokines into groups according to their function

  1. There is an important clinical manifestation that is necessary to be included in the review, to provide a holistic view of the disease. It is necessary to focus part of the discussion on how population diversity impacts acute respiratory distress syndrome. Some questions that are important to address are:
  • Can these differences in mortality rate be attributed to differences in populations?
  • Is there any difference in race-ethnicity between studies?
  • What is the average age of the individuals from one study vs the other?
  • In other words, a more rigorous comparison among studies is necessary so the results can be compared.

 An overview of the most important factors affecting the development of CS, such as age, sex, race (including the outflow of vitamin D deficiency, which is also correlated with race), and obesity, has been added to the text. The inclusion of population characteristics proved to be very important and improved the content of the paper.

  1. Another important aspect that must be included in the review is the key role of sex and age on cytokine synthesis after COVID-19 infection (https://www.ncbi.nlm.nih.gov/pmc/articles/PMC7761414/).

There is a significant difference in the way males and females face COVID-19 infection and it has to do with the proportion of cytokines and chemokines. I strongly suggest reading about these differences because this can help you explain the differences in mortality rates reported in other articles.

 Suggested comments have been added to the manuscript.

Again, we would like to thank you for your time, expertise, and effort in correcting our paper, which improved due to the changes you have proposed. We hope that now it fulfills the requirements to be published.

Kind regards,

Paulina Niedźwiedzka-Rystwej

Round 2

Reviewer 3 Report

Revision of version 2 of the article:

Immune signature of COVID-19 – in-depth reasons and consequences of the cytokine storm.

The article is much more complete now, only one more aspect should be added.

There is no (or at least, I couldn’t find it) clear explanation for the categories mild, moderate and severe.

It is necessary to explain the origin of these categories, if they are clinical categories, or not and on what parameters they are based on. It is necessary to explain better.

In this sense, if the categories are better explained, then In figure 2 the “moderate” condition could be added as a third domain (circle), which will nourish the article and allow an even more interesting discussion.

Author Response

Dear Reviewer,

Thank you for noticing the improvement in our paper after your kind suggestions. Again, we would like to ask you to review the corrections we made after your reading.

In the introduction we have added a paragraph explaining the three different severity states of COVID-19. We have described what the symptoms are and hope that it is now clearer. Furthermore, in the section on signalling pathways, we have highlighted there is a lack of research on cytokine profiles in the different COVID-19 severity states and why we have focused on mild and severe. Therefore, we do believe that there is no data (or we could find it, sorry) on moderate condition of COVID-19 in the sense of cytokine storm (only one report was found to stay there are no differences between the profiles of cytokines according to these categories, including moderate: https://www.immunology.ox.ac.uk/covid-19/covid-19-immunology-literature-reviews/cytokine-profile-in-plasma-of-severe-covid-19-does-not-differ-from-ards-and-sepsis others discuss only the mild and severe category. Summing up, due to the reasons above we are sorry but we were unable to provide changes in the Figure 2.

We do hope that the explanations made in the text will fulfil the subject.

Kind regards,

Paulina Niedźwiedzka-Rystwej